# Torquetenovirus Viremia Quantification Using Real-Time PCR Developed on a Fully Automated, Random-Access Platform

**DOI:** 10.3390/v16060963

**Published:** 2024-06-15

**Authors:** Pietro Giorgio Spezia, Fabrizio Carletti, Federica Novazzi, Eliana Specchiarello, Angelo Genoni, Francesca Drago Ferrante, Claudia Minosse, Giulia Matusali, Nicasio Mancini, Daniele Focosi, Guido Antonelli, Enrico Girardi, Fabrizio Maggi

**Affiliations:** 1Laboratory of Virology, National Institute for Infectious Diseases Lazzaro Spallanzani—IRCCS, 00149 Rome, Italy; pietro.spezia@inmi.it (P.G.S.); eliana.specchiarello@inmi.it (E.S.); claudia.minosse@inmi.it (C.M.); giulia.matusali@inmi.it (G.M.); fabrizio.maggi@inmi.it (F.M.); 2Laboratory of Medical Microbiology and Virology, Department of Medicine and Technological Innovation, Italy; Ospedale di Circolo e Fondazione Macchi, University of Insubria, 21100 Varese, Italy; federica.novazzi@uninsubria.it (F.N.); angelopaolo.genoni@uninsubria.it (A.G.); francy.dragoferrante@gmail.com (F.D.F.); nicasio.mancini@uninsubria.it (N.M.); 3North-Western Tuscany Blood Bank, Pisa University Hospital, 56124 Pisa, Italy; daniele.focosi@gmail.com; 4Laboratory of Microbiology and Virology, Department of Molecular Medicine, Sapienza University of Rome, 00185 Rome, Italy; guido.antonelli@uniroma1.it; 5Scientific Direction, National Institute for Infectious Diseases Lazzaro Spallanzani—IRCCS, 00149 Rome, Italy; enrico.girardi@inmi.it

**Keywords:** TTV, real-time PCR, Hologic Panther Fusion^®^, open channel

## Abstract

Quantification of Torquetenovirus (TTV) viremia is becoming important for evaluating the status of the immune system in solid organ transplant recipients, monitoring the appearance of post-transplant complications, and controlling the efficacy of maintenance immunosuppressive therapy. Thus, diagnostic approaches able to scale up TTV quantification are needed. Here, we report on the development and validation of a real-time PCR assay for TTV quantification on the Hologic Panther Fusion^®^ System by utilizing its open-access channel. The manual real-time PCR previously developed in our laboratories was optimized to detect TTV DNA on the Hologic Panther Fusion^®^ System. The assay was validated using clinical samples. The automated TTV assay has a limit of detection of 1.6 log copies per ml of serum. Using 112 samples previously tested via manual real-time PCR, the concordance in TTV detection was 93% between the assays. When the TTV levels were compared, the overall agreement between the methods, as assessed using Passing–Bablok linear regression and Bland–Altman analyses, was excellent. In summary, we validated a highly sensitive and accurate method for the diagnostic use of TTV quantification on a fully automated Hologic Panther Fusion^®^ System. This will greatly improve the turnaround time for TTV testing and better support the laboratory diagnosis of this new viral biomarker.

## 1. Introduction

Over recent years, Torquetenovirus (TTV) and related anelloviruses have been the subject of intense research [1,2,3]. First identified using molecular techniques in the blood of a patient with post-transfusion transaminitis but negative for classic hepatitis viruses [4], TTV is a ubiquitous non-pathogenic virus, known to be the prototype of a large group of genetically related but distinct circular, single-stranded DNA viruses able to infect humans and various animals and currently classified into the family Anelloviridae [5,6,7,8]. The anelloviruses are characterized by an extremely high degree of genetic heterogeneity, which has led us to recognize TTV and at least two other smaller viruses, Torque teno mini virus (TTMV) and Torque teno midi virus (TTMDV). Each of these viruses consists of numerous species and strains. The recent evidence that TTV represents the most abundant component of the human virome and establishes a highly successful interaction with the infected host [1] has highlighted the importance of using the monitoring of TTV viremia as a biomarker of functional immune competence [9,10,11,12,13,14]. Thus, the use of diagnostic tests that can be performed for rapid and sensitive quantification of TTV is of utmost importance for follow-up monitoring and therapy decisions in several patient settings, notably solid organ transplant recipients [15,16].

For a long time, molecular diagnosis of TTV has been performed using in-house-developed PCR assays, mostly designed based on the highly conserved untranslated region (UTR) of the viral genome [17,18]. PCR offers high specificity and sensitivity in TTV detection, but routine workflows are time-consuming and require significant hands-on time for nucleic acid extraction and sample preparation. Recently, a commercial assay has been developed and is widely used, simplifying the intra- and inter-laboratory standardization of the generated quantitative results [18,19].

However, the increased need for testing TTV in an ever-larger number of patients makes the development of better-performing diagnostic systems urgent. Automated random-access platforms have been created to combine the benefits of PCR analysis with on-demand sample processing, resulting in less hands-on time and a shorter overall turnaround time. Notably, some of these platforms allow the installation of self-designed PCRs to expand the existing portfolio to pathogens that are not tested routinely.

We, therefore, sought to develop a fully automatized real-time PCR for TTV quantification on the Hologic Panther Fusion^®^ System by utilizing its open-access function.

## 2. Materials and Methods

### 2.1. Manual TTV Real-Time PCRs

TTV presence and loads were evaluated by using a single-step universal TaqMan^®^ real-time PCR assay as previously described [17,18]. The assay is based on a highly conserved fragment of the viral UTR, and it amplifies a 63-nucleotides length amplicon by using the following primers and probes: forward primer AMTS 5′-GTGCCGIAGGTGAGTTTA-3′, position nucleotides 177 to 194; reverse primer AMTAS 5′-AGCCCGGCCAGTCC-3′, position nucleotides 226 to 239; and probe AMTPTU 5′-TCAAGGGGCAATTCGGGCT-3′, position nucleotides 205 to 223. Nucleotide (nt) positions are according to isolate NCBI Reference Sequence NC_002076. The assay has the potential for sensitive and specific detection of all the species of TTV designated so far, but it does not detect TTMV and TTMDV, as determined via sequence data analysis and confirmed by testing plasmid sequences. The procedures used for the quantification of copy numbers and the evaluation of specificity, sensitivity, intra- and inter-assay precision, and reproducibility have been previously described [17,18]. The lower limit of sensitivity is 1.0 log DNA copies/mL in both plasma and serum.

The commercial TTV R-GENE kit (BioMerieux, Marcy-l’Étoile, Lyon, France) was the second rtPCR assay used. This assay can find and measure TTV DNA in plasma and whole blood samples by using the TaqMan 5′nuclease technology, and it enables the amplification of a UTR fragment with 128 base pairs. The TTV R-GENE kit includes extraction and inhibition of internal control, as well as four quantification standards.

### 2.2. Clinical Samples and Positive Controls

A total of 112 randomly selected serum samples were studied. The samples were obtained from patients undergoing routine virological analyses. All the samples were tested using manual rtPCR methods that are currently used for TTV diagnosis, and the results obtained (83 samples were positive and 29 samples were negative) were highly concordant between the methods. Anonymized aliquots were immediately prepared, stored, and kept under sterile conditions at –80 °C until use. Informed consent was obtained from each recruited patient. 

A positive control plasmid was cloned using the in-house real-time PCR target fragment, previously amplified via nested PCR by using AMTS and AMTAS as inner primers, into a pCR 2.1 vector (Invitrogen, Carlsbad, CA, USA). Plasmid concentration was analyzed using NanoDrop One (Thermo Fisher Scientific, Waltham, MA, USA) according to the manufacturer’s instructions and used as a standard template diluted in serum from 1.0 log to 7.0 log DNA copies per reaction. This study was approved by the ethics committee with protocol study number 61-2023, 4 October 2023.

### 2.3. Panther Fusion TTV Real-Time PCR

TTV real-time PCR was performed on the fully automated Panther Fusion^®^ molecular system (Hologic). A total of 250 µL of the sample (serum-diluted plasmid or clinical samples) was diluted in 250 µL of specimen diluent buffer, transferred into a specimen lysis tube containing 710 µL of lysis buffer, and then loaded onto the instrument. The system used 360 µL of this mixture for nucleic acid extraction using the Panther Fusion^®^ Extraction Reagent-S^®^, which contained an internal control. Extracted nucleic acid was eluted in a final volume of 50 µL, and 5 µL of eluate was used in 25 µL RT-PCR reactions in the Open Access RNA/DNA Enzyme Cartridge (Hologic). PCR reactions with varying final concentrations of MgCl_2_ (2.5 mM vs. 4 mM) and the TTV (0.6 µM vs. 1 µM) primer and probes (0.4 µM vs. 0.3 µM) concentrations were run in parallel to determine the optimal PCR conditions. 

### 2.4. Analytical Performance Evaluation 

The performance of linearity was assessed with a 10-fold dilution series of a serum TTV-positive clinical sample. Dilutions were tested in triplicate. The analysis of the limit of detection (LoD) was assessed by testing a negative serum TTV sample spiked with the positive control plasmid at decreasing concentrations. LoD was calculated using probit-analysis. The repeatability and calculation of intra-assay and inter-assay variations were evaluated utilizing TTV internal quality control serum samples (IQC) prepared from the QCMD at a concentration of 4.0 log copies/mL. Each sample underwent testing in 10 replicates across two independent experimental runs. Inter-assay variation was assessed using two different reagent batches in separate experimental sessions. Data analysis included calculating coefficients of variation (CVs) for both intra-assay and inter-assay variations.

### 2.5. Statistical Analyses

SPSS software version 25 (IBM, Chicago, IL, USA) was used for statistical analysis. A transformed TTV load in log format was used for analysis. The Spearman rho coefficient was used to measure the overall correlation between methods. The Bland–Altman analysis was used to analyze the concordance and mean differences between assays. Overall agreement between assays was measured using weighted Cohen’s kappa (κ). Tests were two-sided, and *p* < 0.05 was considered statistically significant. Passing–Bablok regression analysis was employed to assess regression and compare methods. To evaluate analytical efficiency by comparing methods, values above the LoD within the analytical range were considered.

## 3. Results

### 3.1. Optimization of PCR Parameters

The optimized final concentrations of the PCR reaction components and thermal cycling conditions are shown in Table 1. Samples were defined as TTV positive or TTV negative based on the following defined analysis parameters: analysis start cycle of 10 cycles; baseline correction slope limit of 50 in the FAM channel; and cycle threshold (Ct) of 1331 RFU. The efficiency, as defined by the calibration curve, for the TTV assays (with a Ct range of 15.0–30.0; 10.2—2.2 log copies/mL), was 117%.

### 3.2. Analytical Sensitivity 

The analytical sensitivity of the Panther Fusion^®^ TTV real-time PCR method was determined using serial dilutions of TTV-negative plasma matrix spiked-in with TTV plasmid with 20 replicates each. The lowest concentration with a 95% hit rate was considered the estimated limit of detection. As a result, the claimed limit of detection was 42 copies/mL (1.6 log copies/mL). The TTV assay was able to detect all dilution replicates of the plasmid down to 85 copies/mL (1.9 log copies/mL) and 95% of the replicates at 42 copies/mL (1.6 log copies/mL) (Table 2).

### 3.3. Assessment of Intra- and Inter-Run Variabilities

Both intra-run (repeatability) and inter-run (reproducibility) variabilities were assessed using the QCMD-designed IQC panel tailored for TTV analysis. Intra-run variability was assessed by performing 10 repeated measurements on two different samples from the IQC panel, each containing a level of 4.0 log copies/mL, which represents an intermediate value in the analytical range scale of the assay. Inter-run variability was evaluated by analyzing two different samples from the IQC panel in two independent runs. The percentage coefficient of variation (CV) for intra-run and inter-run variabilities ranged from 1.47% to 2% (range copy number: 11,616 to 24,134) and 1.73%, respectively (range copy number: 10,224–24,134)

### 3.4. Comparison of TTV Detection and Quantification of Clinical Samples 

After establishing the performance of the automatic TTV Hologic Panther Fusion^®^ assay, 112 biological samples were tested for TTV presence and loads, and the results were compared with those obtained using two manual rtPCR methods that are currently used for TTV diagnosis. These two latter methods were highly concordant in terms of qualitative and quantitative results, as already demonstrated [18]. For this reason, the results obtained by these two assays were merged, and the mean TTV load of each sample calculated based on the quantitative values measured using the two PCRs was used for the following comparisons. Considering all the 112 samples, 75 (70%) and 83 (74%) were positive using the automatic TTV assay and manual TTV assay, respectively. In detail, of the 75 samples that were positive using both TTV assays, 66 (88%) were quantitatively concordant, having a viral load within the range of linearity of the assays, with nine (12%) samples being TTV positive but detected below the lower limit of quantification of the automatic TTV assay and within the range of linearity using the manual assay. Twenty-nine of the 112 (26%) samples were concordantly TTV negative using both assays. On the contrary, eight samples (7%) were discordant, resulting in TTV positivity in only one assay (Table 3). Overall, the assay concordance was 93%, with excellent agreement in TTV detection, as revealed by a Cohen’s κ value of 0.83. 

Loads of TTV DNA from the samples with concordant results were examined. Among the 66 samples with a viral load measurable within the range of linearity of both assays, the median load was 3.9 log copies per ml (range: 1.9–9.8 log copies/mL), as determined using the automatic TTV assay. For these samples, the overall correlation between results, as assessed using Spearman rho correlation analysis, was outstanding (*r* = 0.986, *p <* 0.0001, 95% CI: 0.977 to 0.991) (Figure 1). Again, the overall concordance between the results, as assessed using Passing–Bablok linear regression analysis, was excellent (regression linear equation: y = −0.165222 + 1.017393x; 95% CI for intercept −0.5242 to 0.1740 and for slope 0.9683 to 1.0623; Cusum test for linearity *p* = 0.63) (Figure 2). The Bland–Altman analysis showed a mean difference between the two methods of 0.06 log copies/mL with a 95% CI of agreement of −0.04 and 0.16 (Figure 3), thus revealing a very similar mean quantification between the assays. A Wilcoxon signed-rank test confirmed that the quantification differences between the two methods were not significant (*p* = 0.23).

Finally, TTV loads from eight discordant blood samples were also examined. These samples, which were negative when tested using the automatic TTV assay, were successfully quantified using the manual TTV assay and showed a mean TTV load of 2.2 log copies/mL (95% CI: 1.9–2.4 log copies/mL), which is extremely low.

## 4. Discussion

Although many aspects of the biology of TTV remain to be defined, convincing evidence exists that TTV replication is associated with the functionality of the immune system and that an immune imbalance can significantly increase the TTV load in blood samples [5,20,21]. Thus, TTV viremia quantification is becoming an important parameter for clinicians in tailor-making maintenance immunosuppression [22,23,24]. The measurement of TTV viremia allows more appropriate and personalized use of immunosuppressive therapy and reduces the risk of post-transplant complications (i.e., infectious complications and graft rejection).

Many recent studies have been focused on determining the best cutoffs of TTV viremia that can be used to predict post-transplant risks of opportunistic infection or graft rejection, thus making the ability to precisely measure TTV load the most important challenge of the different molecular methods used for virus quantification [19,25,26]. TTV cutoffs calculated using different TTV assays have been compared, and TTV-specific independent external quality controls have been implemented to make the comparability of quantitative results from laboratory to laboratory more precise. All this strongly reduces methodological differences in the measurement of TTV and improves interlaboratory reproducibility and reliability.

Based on the high number of transplantations performed worldwide per year and considering that post-transplant infections are the second most common cause of death following transplantation, while rejection is the leading cause of graft loss [27,28], it is easy to hypothesize a larger use of TTV quantification in the next few years. Therefore, it is clear that another crucial challenge is the measurement of TTV viremia using molecular methods developed on fully automated random-access platforms. The use of automated TTV assays can give many advantages, including a reduction in workload, less time consumption per sample analysis, a greater number of TTV tests performed in less time, decreased chances of human errors, and high accuracy and reproducibility of the results. To date, to the best of our knowledge, no automated assay for the qualitative and quantitative detection of TTV viremia in blood samples has been developed.

In this study, we developed an automatized real-time TTV PCR assay using the open channel of the Hologic Panther Fusion^®^ instrument and compared the performance of the assay with those of two manual PCR methods (the commercial TTV R-GENE^®^ assay by BioMerieux and the in-house real-time PCR previously developed in our laboratories) using a panel of clinical samples. All the methods rely on real-time PCR technology and are designed for the highly conserved UTR of the TTV genome [18]. Two methods (i.e., the manual in-house real-time TTV PCR assay and the automated real-time TTV PCR assay) provide a measure of the amount of viral DNA present in a sample by using the same TTV primers and probe.

Overall, some considerations are suggested based on the results of this study. The first is that the developed automated TTV assay demonstrates optimal performance that is strongly comparable in terms of sensitivity and reproducibility with the commonly used manual assays. This makes the assay potentially applicable to the routine diagnosis of TTV infection in various settings of the population and when the need exists to test a very large number of samples. The comparison of qualitative PCR results revealed an overall observed agreement higher than 90%, thus indicating strong assay reliability. Again, when the numbers of TTV DNA copies detected were compared, a strong correlation in the ability to quantify the virus among the assays was found. Although a highly significant correlation between the PCR assays was found, some discrepancies were observed.

Eight samples gave a TTV-positive result using manual PCR and a negative result using automated PCR. Some discrepancies may be attributed to differing input volumes required by the optimized system, particularly for samples with low viral loads. Calculation of the mean levels of TTV DNA showed that these discordant samples had a very low TTV load (2.2 log copies/mL), thus suggesting the small amount of viral DNA present in the clinical sample as the possible cause of this discrepancy. It is of interest to note that the performance of the automated assay was investigated on a panel of TTV loads ranging from 1.2 to 9.8 log copies/mL, which includes very low values of TTV and is more prone to the sensitivity of the PCR method used.

In conclusion, the present study is an attempt toward the application of the measurement of TTV viremia on a fully automated random-access platform in an epoch when the importance of TTV measurement is rapidly increasing. More research is needed to increase the number of biological samples and different species of TTV tested, standardize and compare the results across laboratories, and validate these results using other international standards other than QCMD’s.

## Figures and Tables

**Figure 1 viruses-16-00963-f001:**
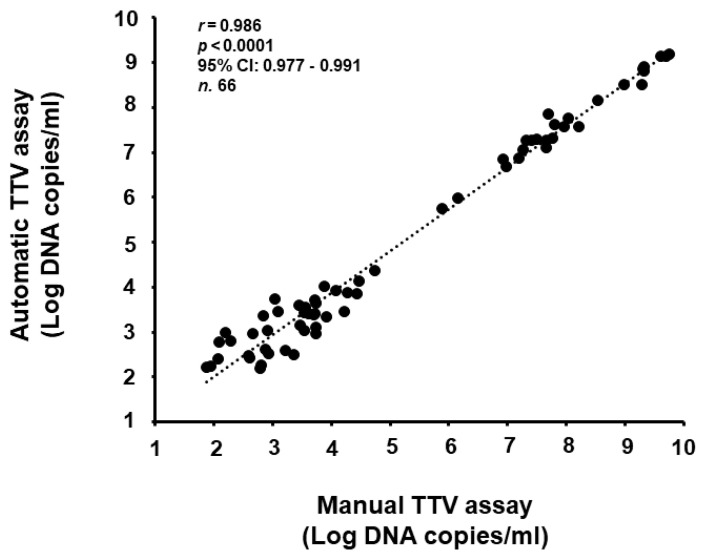
Correlation in TTV levels between the assays evaluated using Spearman’s rho correlation analysis.

**Figure 2 viruses-16-00963-f002:**
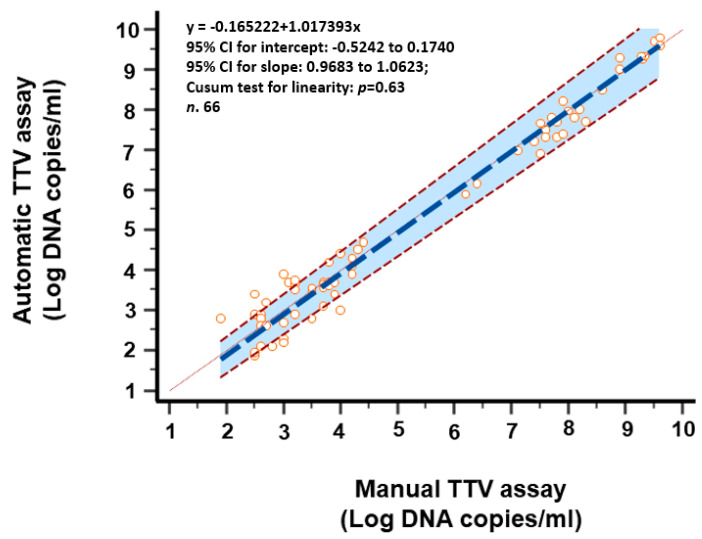
Linear regression analysis of TTV levels of 66 paired quantified samples using Passing–Bablok regression analysis. Shaded areas represent 95% confidence interval limits.

**Figure 3 viruses-16-00963-f003:**
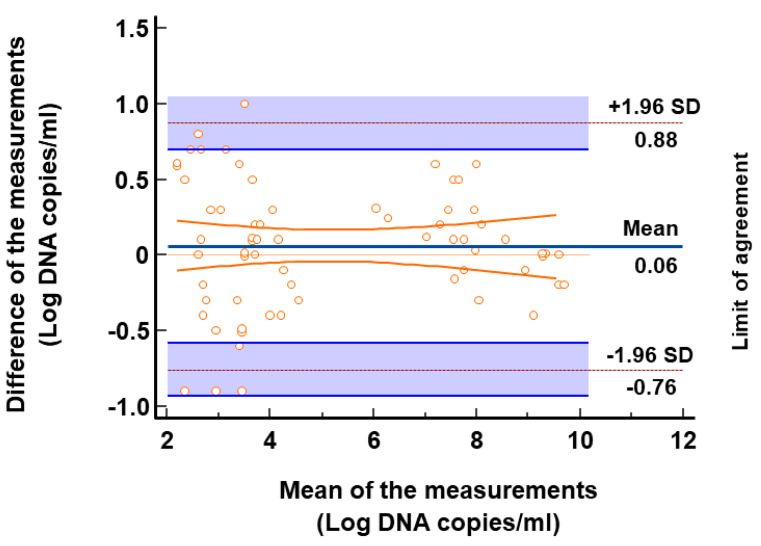
Differences in TTV levels between the assays evaluated using the Bland–Altman plot. Horizontal lines are drawn at the mean difference and the upper and lower limits of agreement. Orange lines and shaded areas represent confidence interval limits for mean and agreement limits, respectively.

**Table 1 viruses-16-00963-t001:** Optimized concentrations of PCR reagents and PCR cycling conditions.

PCR Reagents Concentration	TTV
Water	–
1 M KCl (Hologic^®^)	50 mM
1 M MgCl_2_ (Hologic^®^)	4 mM
1 M Tris, pH 8.0 (Hologic^®^)	10 mM
Forward Primer	1 µM
Reverse Primer	1 µM
Probe	0.3 µM
Internal Control Primer (Hologic^®^)	0.6 µM
Internal Control Probe (Hologic^®^)	0.6 µM
**PCR cycling conditions**	**Temperature—Time**
1	95 °C—2 min
2	95 °C—8 s
3	55 °C—28 s
4	Repeat steps 2–3 for a total of 45 cycles

**Table 2 viruses-16-00963-t002:** Probit analysis.

TTV DNA Plasmid Concentrations (Log Copies/mL)	No. Detected/Replicates	% Detected
10.2	20/20	100%
9.2	20/20	100%
8.2	20/20	100%
7.2	20/20	100%
6.2	20/20	100%
5.2	20/20	100%
4.2	20/20	100%
3.2	20/20	100%
2.2	20/20	100%
1.9	20/20	100%
1.6	19/20	95%
1.3	12/20	60%
1.0	9/20	45%
0.7	5/20	25%
0.4	0/20	0%

**Table 3 viruses-16-00963-t003:** Concordance of TTV load results between the TTV assays.

Manual TTV Assay (Mean of 2 Manual rt-PCR Assays)	No. Examined	TTV DNA with Automatic TTV Assay	
		Negative (No./Percentage)	Positive (No./Percentage)
Negative	29	29 (100)	0 (0)
Positive	83	8 (10)	75 (90)

## Data Availability

Data will be available upon reasonable request to the corresponding author.

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
