# Peer review of "Torquetenovirus Viremia Quantification Using Real-Time PCR Developed on a Fully Automated, Random-Access Platform"

_viruses, 2024, doi:10.3390/v16060963_

Round 1

Reviewer 1 Report

Comments and Suggestions for Authors

The paper by PG Spezia et al., describes the implementation of a quantitative TTV assay on the Hologic Panther platform, which is a random-access and fully automated system. The paper is clearly written and easy to follow. However, some remarks need to be made.

Remarks:

-mention in the introduction that TTV is a DNA virus and also non-pathogenic. Therefore it can be used as a biomarker.

-line 71, explain TTMV and TTMDV. Why are these viruses tested and what is their origin.

-line 75. LLOD of plasma and serum. What was the input and is serum and plasma equally sensitive. Actually, despite the LLOD is 1 log DNA copies/ml (plasma?), it is analytical but clinically not that relevant.

-line 77. I was surprised that the number of randomly selected serum samples was rather low (112). And mention how many positive/negative by which method of selection (is mentioned later in manuscript).

-line 89. Is the method equally used for the positive control method as well as for clinical samples on the Hologic instrument? If so, is the plasmid diluted in serum? Pretreatment would be of interest. Also data on performance isolation of plasma and clinical serum samples are not clearly mentioned.

-line 88. If I understand correctly, this is only used for the serum sample, right (not the plasmid? I am not sure what is dome on what instrument with which method). So 250 ul samples is diluted with 250 plus 710 ul, of which 360 ul is used by the instrument and only 1/10 is used in the assay as eluate. So one can calculate back what could be the LLOD if 5 copies/reaction give a 100% hit rate.

-line 100. This describes the linearity of the assay using 10-fold dilutions of a positive serum clinical sample. At the same time the LOD was tested by adding dilutions of  plasmid in negative TTV samples (plasma line 130?). But he question is whether extraction of a plasmid in plasma and clinically TTV positive serum samples, are equally efficient extracted on the instrument. This is confusing and not clearly explained. I would use a digital PCR method on the clinical sample to get more information on quantification.

-line 105 prepared by QCMD (quality control of molecular diagnostics, Glasgow Scotland).. Are these control plasmid or clinically samples derived?

-line 129. Do you have data on detecting plasmid with plasma or just naked plasmid, to determine the efficiency of quantification by extraction yes/no. Are they equally efficient and if so, these are interesting data.

-line 154. I would make a graph with the quantitative data and showing the differences between the 2 methods. Now it is mentioned that the differences are low copy numbers of below LOD.

-I would remove figure 1; looks always good (even if 2 assays are not that comparable). More interesting is figure 3.

-line 190-193. Can the differences be explained by input volume of the assays used?

-Line 195. The discussion is more focusing on what could be the use of a TTV assay (line 196-222). I would focus on the next steps that need to be developed to implement TTV testing and quantification, i.e. the development of an (international) standard, like using the methodology WHO-reference labs use, of using ddPCR based assays in a multicenter study to start developing a standard. Also mention something about the different genotypes which could be, or not, of importance.

Reviewer 2 Report

Comments and Suggestions for Authors

The method comparison of manual vs automated analysis of TTV in serum is well designed and reported. Primer/probes are correctly reported.

I have a few comments mostly concerning log-transformation.

1. The nomenclature of log-numbers should be universal throughout the manuscript. In abstract the term "log copies" is used (Li22), in Materials and Methods the term log10 (Li74), or 101 to 107 (Li 85). Please use the same nomenclature. I suggest the latter.

2. The limit of sensitivity and detection is somewhat confusing. In the abstract the limit of detection is set at 1.7 log copies per ml of serum (see comment 1 about nomenclature). In Materials and Methods the manual method is referenced to have lower limit of sensitivity of 1.0 log10 DNA copies/ml of plasma or serum. However, when looking at the referenced prior works the limit of sensitivity in ref #17 is 1 x 103 copies per ml. And in ref #18 either 10 copies per µl or 12 copies per ml. Of these numbers the first two are realistic. but in any case they vary from the number noted in Materials and Methods in the present paper. Please revise so that the correct numbers are used both in abstract and the rest of the manuscript.

A quick review of the material used for analysis from 250 µl serum to 5 µL extract in the PCR reveals that potentially 3 % of the original DNA is used as template. Thus the Table 2 Probit analysis seems realistic.  

3. The method comparison is using log10-transformed data (figures 1-3). And the report of intra and inter-rum variability is reported as below 2%, which is extremely impressive in a qPCR analysis. Unless these results are on the log10 data that are not back-transformed. Please report CV% on DNA copy number.

Round 2

Reviewer 1 Report

Comments and Suggestions for Authors

No further comments. The authors responded to the remarks and suggestions.